# Stochastic characterization of navigation strategies in an automated variant of the Barnes maze

Ju-Young Lee[1,2], Dahee Jung[1], Sebastien Royer[1,2]*

[1]Center for Functional Connectomics, Brain Science Institute, Korea Institute of Science and Technology (KIST), Seoul, Republic of Korea; [2]Division of Bio-Medical Science and Technology, KIST School, Korea University of Science and Technology (UST), Seoul, Republic of Korea

**Abstract** Animals can use a repertoire of strategies to navigate in an environment, and it remains an intriguing question how these strategies are selected based on the nature and familiarity of environments. To investigate this question, we developed a fully automated variant of the Barnes maze, characterized by 24 vestibules distributed along the periphery of a circular arena, and monitored the trajectories of mice over 15 days as they learned to navigate towards a goal vestibule from a random start vestibule. We show that the patterns of vestibule visits can be reproduced by the combination of three stochastic processes reminiscent of random, serial, and spatial strategies. The processes randomly selected vestibules based on either uniform (random) or biased (serial and spatial) probability distributions. They closely matched experimental data across a range of statistical distributions characterizing the length, distribution, step size, direction, and stereotypy of vestibule sequences, revealing a shift from random to spatial and serial strategies over time, with a strategy switch occurring approximately every six vestibule visits. Our study provides a novel apparatus and analysis toolset for tracking the repertoire of navigation strategies and demonstrates that a set of stochastic processes can largely account for exploration patterns in the Barnes maze.

*For correspondence:
sebiroyer@gmail.com

Competing interest: The authors declare that no competing interests exist.

## eLife assessment

This study presents a **valuable** new behavioral apparatus aimed at differentiating the strategies animals use to orient themselves in an environment. The evidence supporting the claims is **solid**, with statistical modeling of animal behavior. Overall, this study will attract the interest of researchers exploring spatial learning and memory.

## Introduction

The neural mechanisms underlying navigation strategies have received considerable interest from both experimental and theoretical research (*Nyberg et al., 2022*; *Poucet and Hok, 2017*; *Bicanski and Burgess, 2020*; *Erdem and Hasselmo, 2014*). However, the fact that multiple strategies might be used within the same environment adds another layer of complexity to this problem (*Packard and McGaugh, 1996*; *Berthoz, 2001*; *Arleo and Rondi-Reig, 2007*; *Iglói et al., 2009*; *Chersi and Burgess, 2015*). How navigation strategies are selected is an intriguing question that needs to take into consideration several factors. One important factor is the nature and availability of spatial cues, which can largely vary between environments. Wide open fields such as deserts or city plazas are believed to promote vector-based strategies that rely on distal landmarks and the elaboration of a cognitive map (*Bicanski and Burgess, 2020*; *O'Keefe and Nadel, 1978*; *Bush et al., 2015*;

*Poucet and Hok, 2017*; *Buzsáki and Moser, 2013*), whereas intricate networks of channels/streets should privilege response-egocentric or sequential-egocentric strategies that rely on local cues and memory of left/right turns (*Packard and McGaugh, 1996*; *Save and Poucet, 2000*; *Rondi-Reig et al., 2006*). Another factor is the level of familiarity with the environment. Exploratory behaviors are expected to predominate in novel environments, whereas vector-based navigation and egocentric strategies should progressively increase as animals become more familiar with landmark layouts and routes to goals, respectively. Additionally, the motivational factors driving animal motion in the environment, such as reward-seeking, threat-avoidance, and uncertainty resolving, can also affect navigation strategy (*Packard and McGaugh, 1996*; *Huang et al., 2012*; *Shamash et al., 2021*).

The Barnes maze provides a suitable platform for examining the repertoire of strategies and the shift in strategy that accompanies learning of the environment (*Barnes, 1979*). In this task, rodents explore a circular arena until they find an escape box located under one of several holes distributed along the arena periphery. Simple criteria are commonly used to classify the strategy employed in each trial, which are largely based on the number of visited holes and the assumption that the same strategy is maintained across an entire trial (*Bach et al., 1995*; *Suzuki and Imayoshi, 2017*; *Negrón-Oyarzo et al., 2018*; *Gawel et al., 2019*). However, it is largely unclear whether these criteria are valid and accurately capture the full range of strategies used.

To address these issues, we developed an automated variant of the Barnes maze and performed learning, probe test, and reversal learning experiments commonly carried out in Barnes mazes. Furthermore, we developed new analytical tools to characterize the statistical patterns of vestibule visits; generated vestibule sequences via stochastic processes intuitively compatible with random, serial, and spatial strategies; and demonstrate that a combination of these stochastic processes can largely account for vestibule visit patterns. These amount to a new method for tracking navigation strategies based on the relative weights of stochastic processes, which we used to demonstrate a shift from random to spatial and serial strategies across days and a strategy turnover period of approximately six vestibule visits.

## Results
### Automated maze

We developed a fully automated maze apparatus allowing to randomize the start and goal positions of individual trials without the need for experimenter intervention (*Figure 1a*, *Figure 1—figure supplement 1*). The maze consists of an enclosed arena with an array of 24 doors evenly spaced along the periphery, and two home boxes moving around the arena's perimeter. Start positions are changed by rotating the arena and the home boxes (*Figure 1b*). Furthermore, the arena has a tinted cover that prevents mice from seeing room cues while still allowing for infrared tracking of mouse trajectories.

Our maze is similar to the Barnes maze in that mice explore an open arena and are required to navigate to a goal location chosen among an array of 24 doors evenly spaced along the periphery of the arena. However, unlike typical Barnes mazes where orienting cues are external to the maze and animals start at the center of the maze, our maze features two luminous objects inside the arena as orienting cues and start positions are located at the periphery of the arena.

### Behavioral protocol

For a first block of 15 days (acquisition phase), mice performed 10 trials per day. In each trial, they entered the maze from a randomly chosen start position and navigated to the goal position to consume a water reward. After the 15 days, a probe test was conducted, during which the goal door remained closed and mice were allowed to explore the arena for 2 min. Next, for another block of 4 days (reversal phase), mice performed the same task as in the initial block of 15 days, but the goal position was changed. We recorded behavioral data from 10 male and 9 female mice (C57BL/6, 7–8 weeks old), and since very similar patterns were observed for the two groups, we pooled and analyzed the data together in the main figures (male and female groups are analyzed separately in the supplementary figures).

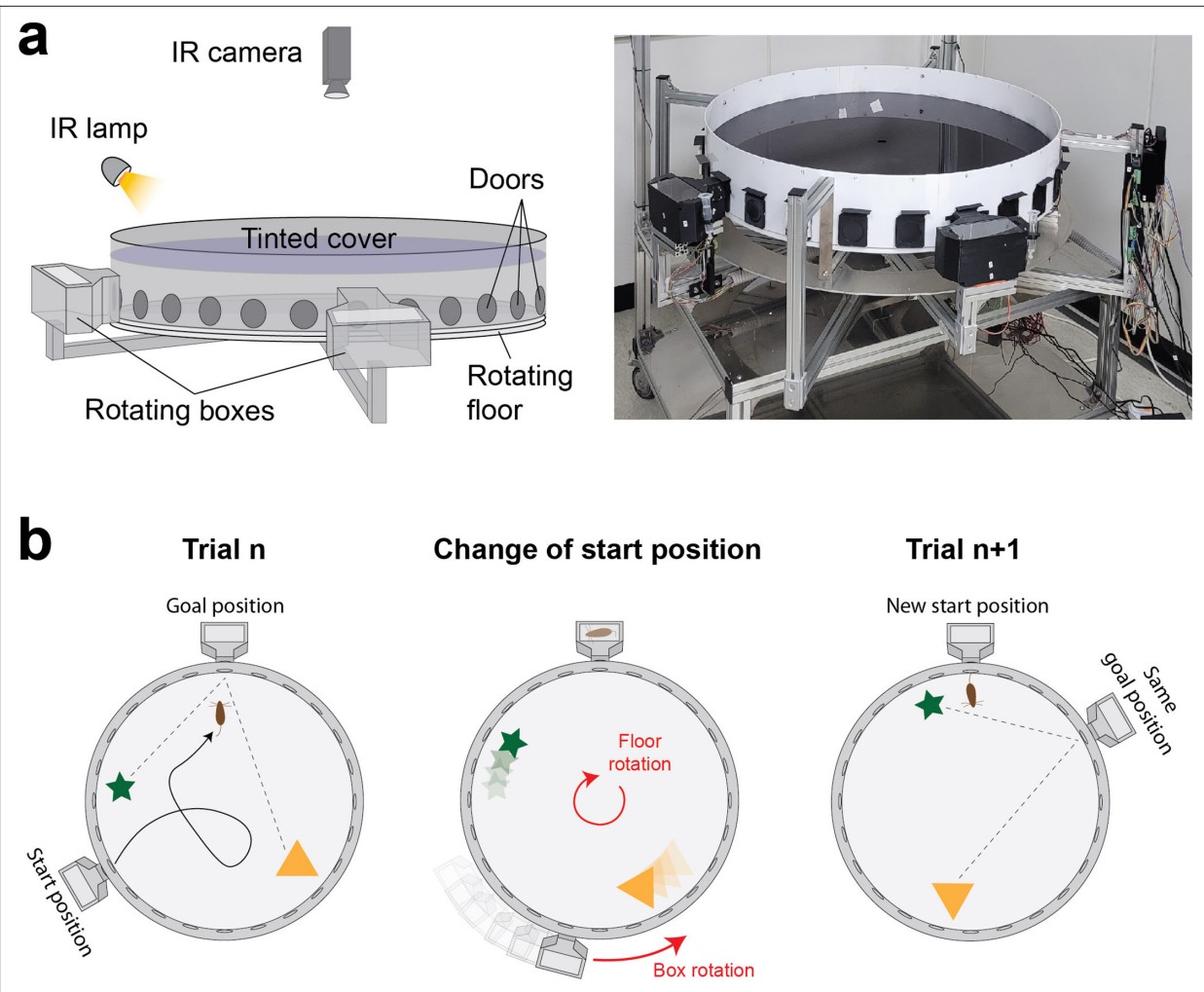

**Figure 1.** Design and operation of the automated variant of the Barnes maze. (**a**) Scheme (left) and picture (right) showing a side view of the apparatus with its different components. The environmental cues are restricted to objects within the arena as the tinted cover blocks visual cues from the surrounding room. (**b**) Scheme showing the operation of the apparatus. A new start position is set by rotating the arena. The goal box is simultaneously moved to align with the goal location. The spatial information remains consistent across trials since the guiding objects are rotated together with the arena.

The online version of this article includes the following figure supplement(s) for figure 1:

**Figure supplement 1.** Detailed view of the automated variant of the Barnes maze.

## Segmentation of spatial trajectories

To analyze the spatial behavior of mice, we decomposed their trajectories into segments that spanned from one vestibule to another, as mice tended to explore the arena and visit multiple vestibules before reaching the goal. To accomplish this, we first detected vestibule visits, and then measured the number of door-intervals each segment spanned, considering the direction of travel associated with the shortest distance. A segment could span from 1 to 12 door-intervals in either the clockwise or counterclockwise direction (*Figure 2a*). Positive door-interval values represented clockwise directions, while negative values represented counterclockwise directions.

## Statistical characteristics of spatial trajectories across days

We analyzed the statistical characteristics of spatial trajectories across days (*Figure 2*, *Figure 2—figure supplements 1–4*). We found that the path length of individual trials decreased across days, with the largest drop observed between day 1 and day 2 (*Figure 2—figure supplement 2a*), consistent with previous reports in the Barnes maze (*Suzuki and Imayoshi, 2017*; *Koopmans et al., 2003*;

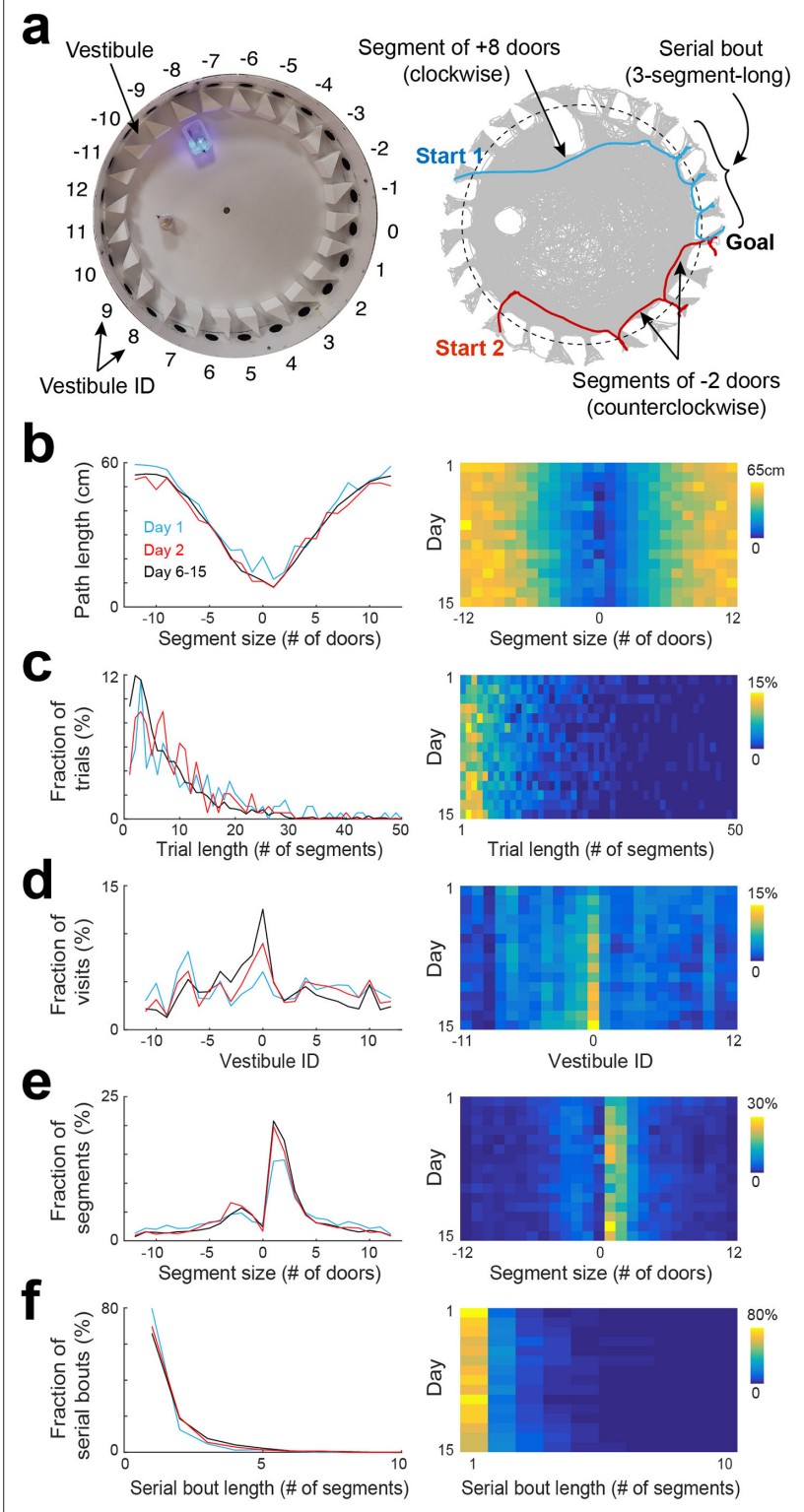

**Figure 2.** Statistical characterization of vestibule sequences across days. (**a**) Top view of the maze (left) and two example trajectories on top of an overlay of multiple trajectories (right) showing vestibules and trajectory segmentation. The goal is at vestibule 0. Running segments span between two vestibule visits. Segment size is defined as the number of door-intervals a segment covers in the clockwise (positive sign) or counterclockwise (negative sign) direction. Serial bouts are bouts of consecutive 1-door-long segments. (**b**) The average path length as a function of segment size for 3 time periods capturing the dynamic range of the distributions (lines) and across

*Figure 2 continued on next page*

*Figure 2 continued*

days (color coded). (**c–f**) Same display as in (**b**) for the distribution of trial lengths (the number of segments per trial) (**c**), the distribution of visits across vestibules (**d**) (notice the peak at vestibule 0, which is the goal location), the distribution of segment sizes (**e**), and the distribution of serial bout lengths (**f**).

The online version of this article includes the following figure supplement(s) for figure 2:

**Figure supplement 1.** Statistical characterization of vestibule sequences across days, for male and female separately.

**Figure supplement 2.** Path length and number of segments per trial across days.

**Figure supplement 3.** Quadrant and vestibule preference during the probe test.

**Figure supplement 4.** Vestibule orientation determines the direction of serial behavior.

---

*O'Leary and Brown, 2013*). However, the path length of individual segments did not show substantial change across days when comparing segments of equivalent spans and directions (*Figure 2b*), suggesting that the overall decrease in trial path length was more associated with a decrease in the number of vestibule visits rather than an alteration of running paths. Next, we calculated the trial length (number of segments per trial). We found that the average trial length decreased across days, with the largest drop observed between day 1 and day 2 (*Figure 2—figure supplement 2b*). The distribution of trial lengths showed a peak for short trials followed by a progressive decline toward longer trials (*Figure 2c*). The proportion of short trials became higher across days, at the expense of longer trials, consistent with the average decrease in trial length.

To determine whether mice relied on allocentric spatial information to learn the goal location, we analyzed the proportion of visits to each vestibule, expecting to see an increasing concentration of visits near the goal location over time. Although no preference was observed on day 1, mice progressively showed a preference for the goal vicinity in subsequent days, supporting their use of allocentric cues to locate the goal (*Figure 2d*). To further validate this finding, we measured the time mice spent in each quadrant of the arena and in each vestibule during the probe test. The percentage of time spent in the goal-containing southwest (SW) quadrant was significantly higher than chance ($p=2.54 \times 10^{-4}$, two-sided Wilcoxon sign-rank test) and other quadrants (*Figure 2—figure supplement 3a*, $p<0.001$, two-tailed paired $t$-test), and the percentage of time spent in the goal vestibule was significantly higher than chance ($p=0.009$, two-sided Wilcoxon sign-rank test) and other vestibules (*Figure 2—figure supplement 3b*, $p<0.05$, two-tailed paired $t$-test), confirming that mice relied on allocentric cues to learn the goal location.

In the Barnes maze, mice occasionally exhibit a search strategy called serial, where they visit consecutive holes in series. To determine the extent of this behavior, we calculated the length of segments (the number of door-intervals they spanned; *Figure 2a*). On day 1, there was a high proportion of short segments (1–2 door-intervals), consistent with the use of serial strategy (*Figure 2e*, $F_{2,108}=3.66$, $p=0.03$, two-way ANOVA). This proportion increased over subsequent days, indicating an increased reliance on serial strategy (14.09 ± 5.81 %, 17.50 ± 5.39%, 18.96 ± 6.1% for day 1, day 2, and day 6–15, respectively; $p=0.001$ for day 1 vs day 2, and $p=0.002$ for day 1 vs day 6–15, two-tailed paired $t$-test).

Interestingly, we found that the peak of the segment length distribution was more than three times (3.56±1.13 time, across days) larger for the clockwise than the counterclockwise direction, indicating that the serial strategy was mostly used in the clockwise direction. This was likely due to the vestibules' orientation, which determined mice body orientation as they left a vestibule, matching the clockwise direction. When we switched the orientation of the vestibules, we observed a reversal of the animals' serial visit direction from clockwise to counterclockwise (*Figure 2—figure supplement 4*).

Additionally, we computed the length of serial bouts, defined as the number of consecutive one-door-interval-long segments, and found that the proportion of serial bouts sharply decayed as a function of serial bout length, with 1-segment-long bouts accounting for 82.62 ± 13.16% of the bouts on day 1 (*Figure 2f*). The proportion of longer bouts was increased across days, consistent with an increased usage of serial strategy (17.38 ± 13.16%, 29.34 ± 17.14%, 32.68 ± 11.38% for day 1, day 2, and days 6–15, respectively).

## Statistical characteristics of spatial trajectories across trials

To determine if the pattern of vestibule visits changed across individual trials within a session, we pulled together the data from days 6 to 15 and performed the same analyses as in *Figure 2* for individual trials (*Figure 3*, *Figure 3—figure supplement 1*). We found that changes across trials were not observed for the distribution of trial lengths and vestibule visits (*Figure 3b and c*), but the distributions of segment length and serial bout length showed an evolution across trials that resembled the evolution across days. Specifically, the first trial showed profiles similar to day 1, and later trials showed profiles similar to later days (*Figure 3d and e*). Hence, a similar evolution of serial strategy was observed across days and trials.

## Stochastic processes for random, spatial and serial strategies

To investigate if the observed pattern of vestibule visits could be explained by random, spatial, and serial strategies, we implemented three distinct stochastic processes (*Figure 4a*), performed simulations using same number of mice and trials and the same start positions as in the experiments, and performed the same analyses as *Figure 2* on the simulated data (*Figure 4b*). Each strategy determined which vestibules would be visited next and was recursively run until the goal (vestibule 0) was reached.

For the random strategy, the identity of the next vestibule was randomly picked based on a uniform probability distribution, which produced a relatively uniform distribution of vestibule visits, similar to day 1 of the experiment. For the spatial strategy, the identity of the next vestibule was randomly picked based on a probability distribution that exponentially decayed as a function of goal distance. This strategy reproduced the concentration of vestibule visits near the goal observed in days 2–15, and partially contributed to the peaks in the distribution of segment lengths. The serial strategy could manifest in either the clockwise or counterclockwise direction. The identity of the next vestibule was determined by incrementing the current vestibule number by a certain step, with the sign of the step being positive for the clockwise direction and negative for the counterclockwise direction, and the size of the step obtained by adding a normal random jitter to the value of 1.2 for the clockwise direction and 2 for the counterclockwise direction. To merge both directions into a single process, we randomly assigned the step's sign with an 80% chance of it being positive and a 20% chance of it being negative. This strategy effectively reproduced the distribution of segment lengths.

## Mixture model combining random, spatial, and serial stochastic processes

None of the individual strategies alone could fully reproduce all of the experimental distributions depicted in *Figure 2*. Thus, we implemented a mixture model that integrates the three strategies, selecting them stochastically based on a specific probability distribution and employing them for a defined number (N) of segments (*Figure 5a*). To determine the optimal probability distribution, we conducted simulations for all possible combinations of probabilities, with each probability ranging from 0 to 100% with an increment of 2. The optimal value of N was sought within a range from 1 to 15. For each probability distribution and N value tested, we conducted the same analyses as in *Figure 2* and compared the results with experimental data through mean square error calculations (*Figure 5b*). The optimal probability distribution and N value were identified as those yielding the lowest mean square error. To mitigate overfitting errors, we repeated all simulations and error calculations 10 time and computed averages to obtain the final probability distribution and value of N.

To track random, serial, and spatial strategies over the 15 days, we repeated all the processes described above for each day, using the same sequences of start positions as in the experiments. The mixture model reproduced the general features of experimental data across days (*Figure 5b*, *Figure 5—figure supplement 1*), with the optimal fit being reached for a value of N=6 (*Figure 5c*). These findings demonstrate that the combination of the three stochastic processes could account for the patterns of vestibule visits.

## Evolution of search strategy across days

Using the probabilities obtained from the mixture model, we examined how the proportion of random, spatial and serial strategies evolved across days. Overall, the spatial strategy was increased across days, from $13.4 \pm 4.16\%$ on day 1–$53 \pm 8.34\%$ on day 15 ($P=1.86 \times 10^{-6}$, two-tailed paired *t*-test),

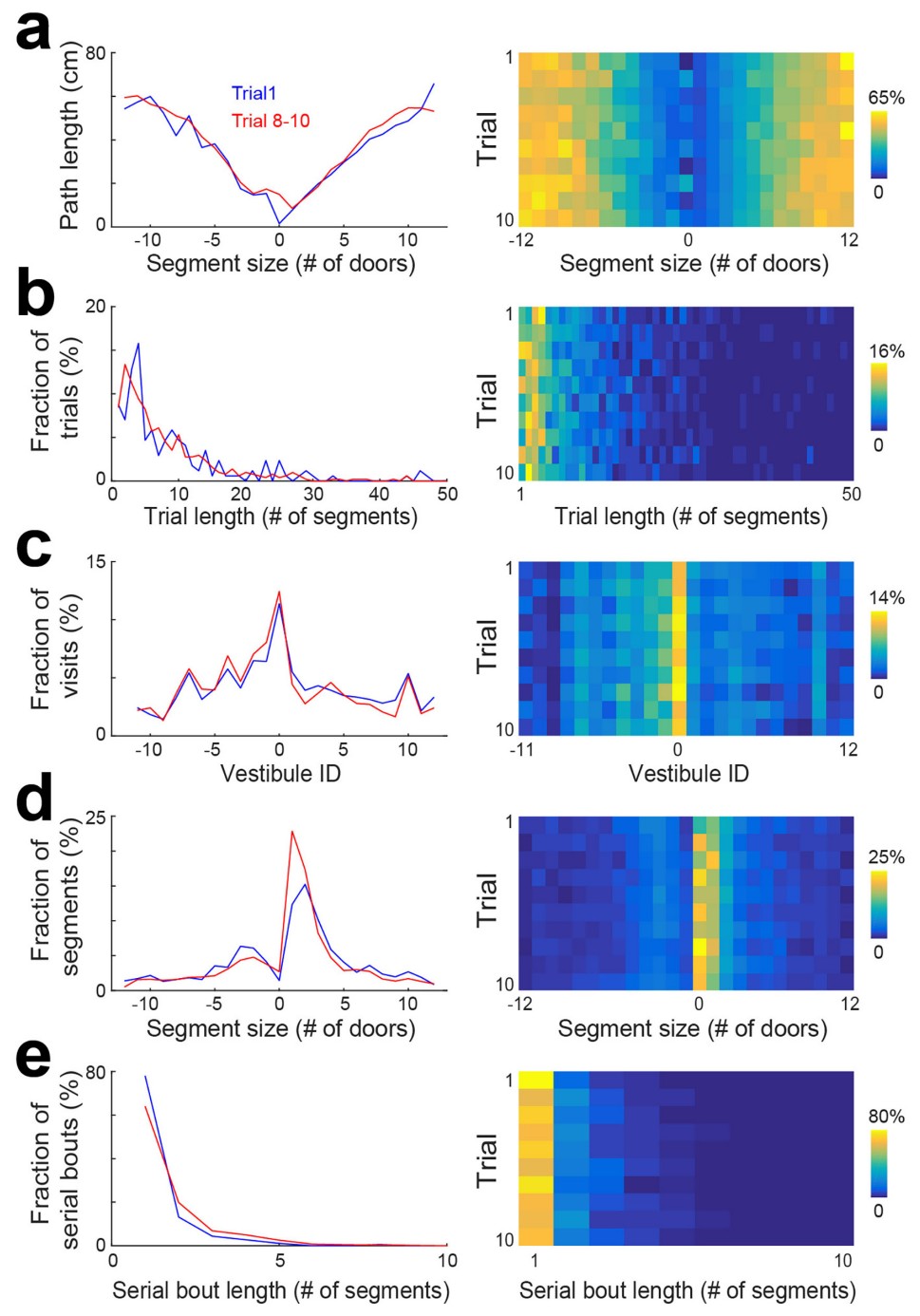

**Figure 3.** Statistical characterization of vestibule sequences across trials. (**a**) The average path length as a function of segment size for 2 time periods capturing the dynamic range of the distributions (lines) and across trials (color coded). The data from days 6 to 15 were used for this analysis. (**b–e**) Same display as in (**a**) for the distribution of trial lengths (the number of segments per trial) (**b**), the distribution of visits across vestibules (**c**), the distribution of segment sizes (**d**), and the distribution of serial bout lengths (**e**).

The online version of this article includes the following figure supplement(s) for figure 3:

**Figure supplement 1.** Statistical characterization of vestibule sequences across trials, for male and female separately.

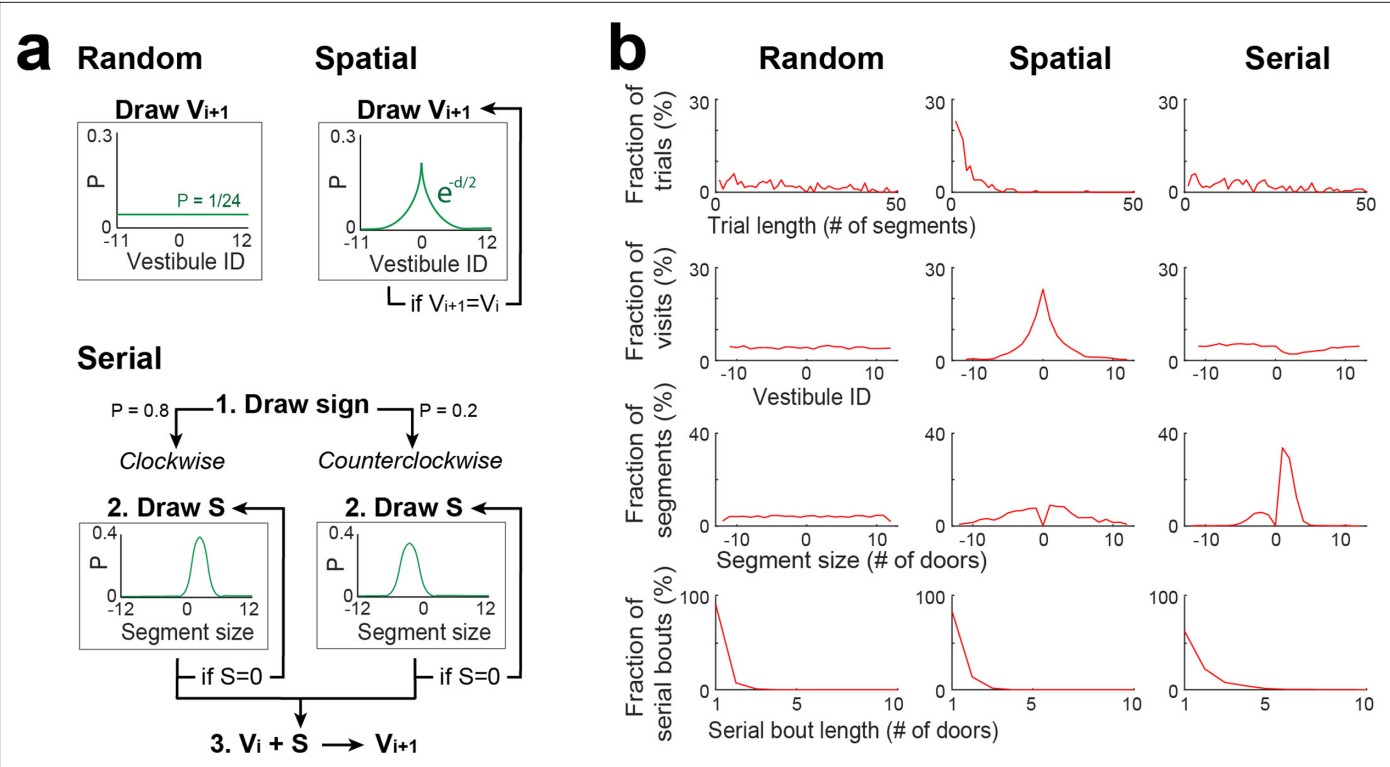

**Figure 4.** Stochastic processes for random, spatial and serial strategies. (**a**) Schemes describing the stochastic processes for random, spatial and serial strategies. Vi +1: the next vestibule visited; Vi: the current vestibule; S: the next segment (a number of door-interval and a sign indicating clockwise/counterclockwise direction); and green lines: the probability distributions for random draws. Random strategy: Vi +1 is randomly drawn from the uniform distribution. Spatial strategy: Vi +1 is randomly drawn from a symmetric exponential distribution, and redrawn if Vi +1 = Vi. Serial strategy: The serial strategy can occur in either the clockwise or counterclockwise direction. The value of S is randomly drawn from a normal distribution with distinct center and width values for clockwise and counterclockwise directions, and is redrawn if S=0. Then Vi is incremented by S to obtain Vi +1. For the implementation of a single process combining both directions, the sign of S is first drawn with a 0.8/0.2 probability bias for positive/negative values, respectively. (**b**) The same analyses as in *Figure 2c–f*, carried on vestibule sequences outputted by each stochastic process over 10 trials and 20 mice. For each trial, the stochastic process was recursively run until Vi = 0. Note that none of the individual processes is reproducing the distributions of *Figure 2*.

largely at the expense of the random strategy, which decreased from 58.2 ± 3.58% on day 1–3.8 ± 4.16% on day 15 (p=6.94 x 10⁻¹¹, two-tailed paired *t*-test), while the serial strategy increased mostly from day 1 to day 2 (28.4 ± 3.1% on day 1 vs 44.6 ± 3.27% on day 2, p=7.62 x 10⁻⁷, two-tailed paired *t*-test; *Figure 5d*, *Figure 5—figure supplement 1*).

To compare our results with previous studies, we also assessed the proportions of random, spatial, and serial strategies using the methods used in those studies (*Figure 5—figure supplement 1d, e*). Specifically, we classified a trial as using a spatial strategy if the number of vestibule visits before reaching the goal was less than 3, a serial strategy if the goal was reached via a serial bout of at least 3 segments, and a random strategy if the number of vestibule visits exceeded 3 and no serial bout led to the goal. Consistent with our mixture model results, we found that the spatial and serial strategies increased across days, while the random strategy decreased. However, the proportions of the random strategy remained relatively high across all days (~46% on day 15), in contrast to the mixture model results.

## Inter-animal variations in search strategy across days

To assess the variability in search strategies across individual animals, we computed the evolution of search strategy for each mouse separately, using the mixture model (*Figure 5—figure supplement 2*). We observed considerable diversity in strategies among individual mice across days, which may partly stem from the lower reliability of model fits for distributions with smaller sample sizes when considering data at the individual mouse level. Nonetheless, discernible strategy preferences persisting

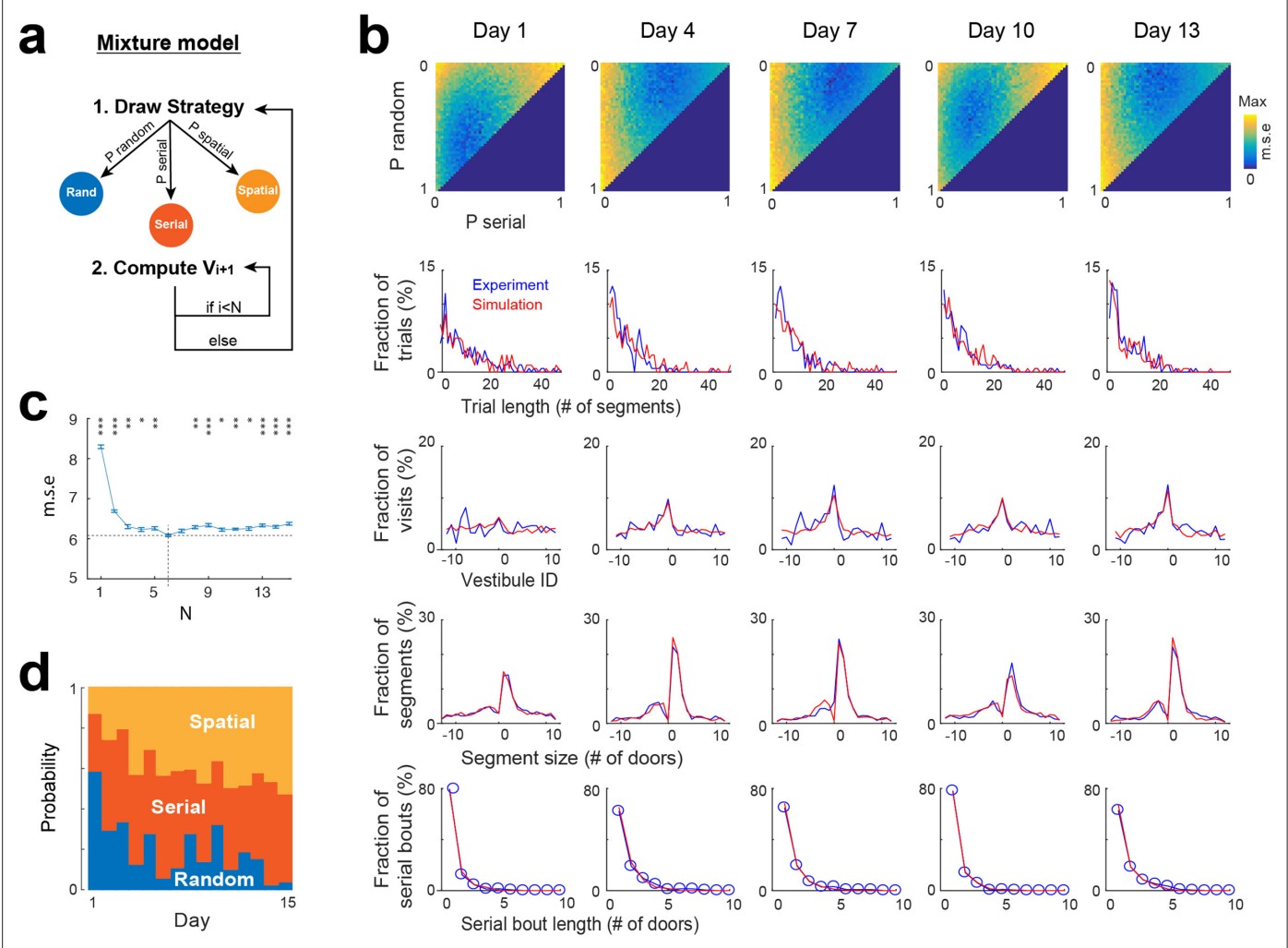

**Figure 5.** Mixture model fitting of strategy evolution across days. (**a**) Mixture model combining the stochastic processes associated with random, spatial and serial strategies. A vestibule sequence is generated in two alternating steps. In step 1, a strategy is drawn according to the set of probabilities P_random, P_serial and P_spatial. In step 2, the stochastic process associated with the selected strategy is used to draw the next N vestibules. The vestibule sequence terminates upon reaching vestibule 0. Start positions are the same as in experiments. (**b**) Fits of experimental distributions for 5 days examples (using N=6). *Color coded*, mean square error (m.s.e) of the fits for all combinations of random, serial and spatial strategies (note that P_spatial = 1 P_random - P_serial). *Line plots*, overlays of experimental (blue) and model (red) distributions, for the best fits. (**c**) Mean square error as a function of N (mean ± s.e.m of 10 simulations for each value of N). A minimum is reached for N=6 (dash lines). *Asterisks*, significance relative to this minimum (*p<0.05, **p<0.005, ***p<0.0005, two-tail unpaired t-test). (**d**) Proportions of each strategy across days, obtained from the best fits.

The online version of this article includes the following figure supplement(s) for figure 5:

**Figure supplement 1.** Strategy evolution across days based on the mixture model fit.

**Figure supplement 2.** Inter-animal variations in search strategy across days.

across days were evident among the animals (*Figure 5—figure supplement 2a*). Furthermore, the average of all individual mouse distributions showed a trend similar to that obtained when pooling data from all mice (*Figure 5—figure supplement 2b*).

Interestingly, we observed significant differences between male and female mice. Initially, male tended to use the random strategy more frequently and the spatial strategy less often than female (*Figure 5—figure supplement 2c*). Moreover, they showed a more pronounced decrease in random strategy across days (male, r=–0.67, p=0.0067, female, r=–0.38, p=0.16, Pearson correlation). Importantly, the progressive increase in spatial strategy was observed in male but not in female mice (male, r=0.895, p<0.0001, female, r=–0.17, p=0.54, Pearson correlation). Conversely, female mice showed

larger increases in serial strategy (male, *r*=–0.25, p=0.36, female, *r*=0.5, p=0.055, Pearson correlation; *Figure 5—figure supplement 2d*).

## Markov chain modeling of the evolution of search strategy within trials

As the task and environment become familiar, it is conceivable that an efficient approach to reach the goal involves a specific sequence of strategies. While the mixture model effectively estimates strategy proportions, it lacks information on the sequence of strategies within trials. To address this aspect, we implemented a Markov chain model where strategies were nodes of a Markov chain and potentially changed after each vestibule visit according to a specific set of transition probabilities, and where vestibules were chosen based on the same stochastic processes as in the mixture model, though the clockwise and counterclockwise directions of the serial strategy were treated as distinct strategies (*Figure 6a*). To estimate the probabilities for both the initial strategy used in a trial and subsequent strategy transitions, we implemented a genetic algorithm that explored the range of possible probability sets and iteratively refined values in order to minimize mean square errors between experimental and simulated distributions (*Figure 6b–d*; see Materials and methods).

To assess how strategy changed within trials, we computed the distributions of segment sizes and vestibule visits for the first 10 segments of the trials, pooling together data from days 6 to 15 (*Figure 6b*). In line with a specific sequence of strategies, these distributions evolved across segments. Notably, the peaks of the distributions were absent in the first segment and gradually emerged in subsequent segments. The Markov chain model successfully reproduced this trend (*Figure 6b*), along with the distributions of trial length and serial bout length (*Figure 6c*). The optimal probabilities for both the initial strategy and strategy transitions remained largely consistent across 10 repetitions of the genetic algorithm (*Figure 6e and f*). Mice typically initiated trials using either the random or spatial strategy (*Figure 6e*). Following the use of the random strategy, they most likely switched to either the serial clockwise or spatial strategy (*Figure 6f*). Both the serial clockwise and spatial strategies represented attractor states, exhibiting a high probability of subsequent reuse (*Figure 6f*).

## Discussion

In this study, we developed an automated variant of the Barnes maze. In general, automated behavioral apparatuses reduce experimenter interference and ensure consistency across time and individuals (*Pioli et al., 2014*; *Zhang et al., 2018*; *Holleman et al., 2019*; *Mei et al., 2020*). To change start positions, the whole arena is rotated to align home boxes with new start positions. Although environmental cues mainly consist of objects within the arena as it is isolated by a tinted cover, it is worth noting the possible influence of external factors such as room sounds and odors, which could introduce conflicting information across trials. Additionally, our maze design may not be conducive to experiments involving wired implants due to the enclosed nature of the arena and boxes. Instead, researchers can explore alternative approaches such as pharmacogenetics and wireless methods for optogenetic and electrophysiological studies. Our maze is similar to the Barnes maze but has distinct features such as vestibules in front of the doors, which made detection of vestibule visits relatively easy, and the use of two luminous objects in the arena for allocentric information. Despite these differences, we found that learning rates and spatial strategies were comparable to those reported in the Barnes maze (*Suzuki and Imayoshi, 2017*; *Koopmans et al., 2003*; *O'Leary and Brown, 2013*). Furthermore, we found that mice had a preferred direction of rotation during the serial strategy, which was determined by the physical configuration of vestibules. Directional preference was observed in several other studies (*Fyhn et al., 2002*; *Stackman et al., 2012*; *Gillani et al., 2014*), which may have also originated from environmental factors.

Our analyses of vestibule visits revealed that mice employed random, spatial, and serial search strategies. While these strategies have been previously considered (*Bach et al., 1995*; *Suzuki and Imayoshi, 2017*; *Negrón-Oyarzo et al., 2018*; *Gawel et al., 2019*), this is the first time that compatible stochastic processes are implemented and shown to complementarily account for exploration patterns. The spatial strategy likely used a form of allocentric navigation, given the distance of the two luminous landmarks from the goal, and may be divided into two phases: first, the learning of the environment and goal location, supported by the formation of a cognitive map in the hippocampal formation (*O'Keefe and Nadel, 1978*; *Ekstrom et al., 2003*; *Grieves and Jeffery, 2017*; *Poulter*

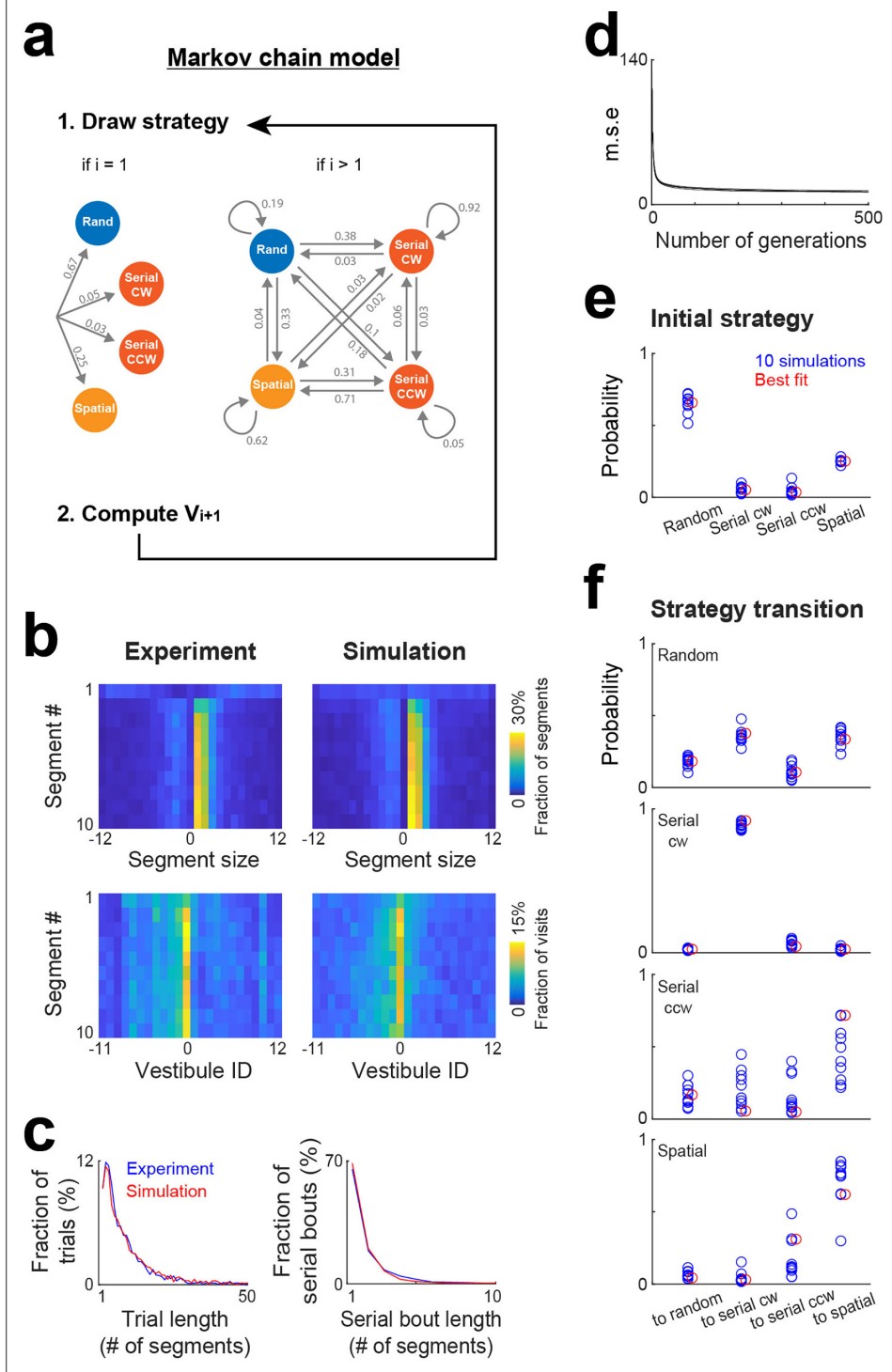

**Figure 6.** Markov chain modeling of within-trial strategy evolution. (**a**) Markov chain model incorporating four strategies (random, serial clockwise, serial counterclockwise and spatial). To generate vestibule sequences, the model iteratively draws a strategy and then a vestibule using the stochastic process associated with the selected strategy, repeating this operation until the goal (vestibule 0) is reached. At the beginning of the trials (i=1), the strategy is drawn according to a set of probabilities determining the initial strategy. Subsequently (i>1), the model transitions between strategies based on another set of probabilities. Numbers indicate the probabilities obtained from the model best fit. (**b**) Experimental and model (best fit) distributions for segment size and vestibule visits, implemented on the level of individual segments for the first 10 segments of the trials. The data from days 6 to 15 were used for this analysis. (**c**) Overlays of experimental (blue) and model (red) distributions for both trial

*Figure 6 continued on next page*

*Figure 6 continued*

length and serial bout length, using the same fit instance as in (**b**). (**d**) Evolution of the mean square error (m.s.e) across generations of the genetic algorithm, for the 10 repetitions of the genetic algorithm. (**e–f**) *Blue circles*, set of probabilities determining the initial strategy (**e**) and strategy transitions (**f**), generated by 10 repetitions of the genetic algorithm. *Red circles*, the probability set that produced the smallest m.s.e.

*et al., 2018*), achieved through competitive learning (*Rolls et al., 2006*; *Si and Treves, 2009*; *Kim et al., 2020*), which provides the substrate for binding place and reward information (*Robinson et al., 2020*; *de Lavilléon et al., 2015*; *Dupret et al., 2010*). Second, the navigation to the goal, which may rely on vector navigation, where goal vectors (*Sarel et al., 2017*; *Ormond and O'Keefe, 2022*) are implemented via the comparison of grid cell codes for goal and current positions (*Bush et al., 2015*; *Banino et al., 2018*; *Boccara et al., 2019*), or on topological navigation, where the trajectory to the goal emerges from signal propagation in the place cell network (*Poucet, 1993*; *Muller et al., 1996*; *Redish and Touretzky, 1998*), and may be assisted by lookahead mechanisms such as theta sweeps and/or ripple-associated replays (*Diba and Buzsáki, 2007*; *Davidson et al., 2009*; *Wikenheiser and Redish, 2015*). The mechanism behind the serial strategy is less clear. The intense and systematic exploration of vestibules in this strategy may stem from a general association between vestibules and rewards. However, the repetitive left turns is reminiscent of egocentric responses supported by the dorsolateral striatum (*Packard and McGaugh, 1996*; *Devan and White, 1999*; *Yin et al., 2004*), and thigmotaxis—an innate behavior enhanced by anxiety, where animals tend to stay close to walls while navigating open environments (*Huang et al., 2012*; *Devan and White, 1999*; *Treit and Fundytus, 1988*). Thus, the serial strategy might involve the combination of innate thigmotaxis-like behavior with associative learnings underlying vestibule-reward and left turn-reward associations. Finally, the random strategy might be related to exploratory behaviors and engage an ensemble of brain regions important for sensory integration, decision-making and reinforcement learning such as the hippocampus, prefrontal cortex, and basal ganglia (*Chersi and Burgess, 2015*; *Roberts et al., 1962*; *Hok et al., 2005*; *Yin et al., 2005*; *Chakravarthy et al., 2010*; *Rinaldi et al., 2020*; *Dong et al., 2021*; *Pernía-Andrade et al., 2021*; *Choi et al., 2021*).

A shift in the proportion of random, spatial and serial strategies was observed across days. Several factors might contribute to this shift, including learning of the environment and goal location, changes in motivation for exploration versus goal-directed navigation, and the evaluation of each strategy's benefit via reinforcement learning. The spatial strategy progressively increased, mostly at the expense of the random strategy. This trend suggests a diminishing interest in exploration and an increasing benefit from employing the spatial strategy as the mice became more familiar with the environment and goal location. Consistent with this hypothesis, the development of the spatial strategy approximately matched the development of spatial maps in the hippocampus (*Kim et al., 2020*) and the growth pattern of hippocampal feedforward inhibitory connectivity (*Ruediger et al., 2011*), both showing progressive increases that reached plateaus after a week. In contrast, the serial strategy showed a sudden increase from day 1 to day 2, indicating that this goal-directed strategy is associated with rapid learning and could already be reinforced on day 2. However, the strategy shift was not uniform across the mouse population, as male and female mice showed distinct trends. Female mice showed no progressive increase in spatial strategy and initially relied more on the spatial strategy while using the random strategy less compared to male mice. This difference might be explained by faster learning of goal location and/or a stronger inclination towards goal-directed navigation over exploration in female mice.

Maintaining the same strategy across several trials is particularly useful for successfully reaching the goal via the serial strategy, and might rely on mPFC working memory and strategy selection functions (*Touzani et al., 2007*; *Schuck et al., 2015*; *Guise and Shapiro, 2017*). Our mixture model and experimental data were optimally matched when the same strategies were maintained for approximately six segments. Most trials comprised fewer than six segments (~60% and~66% of trials on day 1 and 15), thus involved the use of the same strategy throughout the whole trial. However, a substantial number of trials extended beyond six segments, thus involved switching between strategies. This suggests that commonly used methods for categorizing navigation strategy on a trial-by-trial basis may not accurately capture the full repertoire of strategies employed.

Our Markov chain model effectively replicated within-trial evolution of strategies, providing a more comprehensive insight into the probabilistic rules governing strategy selection. Interestingly, mice preferentially initiated trials with either the random or spatial strategy once they had learned the task. One possible explanation is that mice learned, through negative reinforcement, to avoid using the serial strategy at the start of trials, given that start positions were intentionally set to be at least 2 door-intervals away from the goal position. Furthermore, both spatial and serial strategies exhibited a relatively high probability of being reused in subsequent segments, consistent with the observed strategy continuity assessed with the mixture model. While the Markov chain model offered a more comprehensive understanding of strategy selection rules, it was considerably more computationally intensive than the mixture model, requiring up to 20 time more execution time. Therefore, in certain cases, such as tracking strategy repertoire across days, the mixture model might be more practical and preferable due to its computational efficiency.

In conclusion, we have developed a fully automated variant of the Barnes maze along with analytic tools allowing to track the repertoire of navigation strategies over time. Furthermore, we have demonstrated that a set of stochastic processes compatible with random, spatial and serial strategies could largely account for vestibule visit patterns. This achievement opens up future prospects for a comprehensive characterization of animal behavior, extending across various environments, behaviors and species. Our tools might be combined in the future with optogenetic, pharmacogenetic manipulations and/or wireless electrophysiology systems to investigate the neural mechanisms underlying strategy selection, whereas future work in silico might investigate neuronal network models generating exploration patterns equivalent to the stochastic processes.

## Materials and methods

### Experimental subjects

All experiments were conducted in accordance with institutional regulations (Institutional Animal Care and Use Committee of the Korea Institute of Science and Technology) and conformed to the Guide for the Care and Use of Laboratory Animals (NRC 2011). Ten female and ten male mice (C57BL/6, 7–8 weeks old, Jackson Laboratory) were initially used for the experiment. However, one female was excluded from the study due to exhibiting excessive circling and thigmotaxis behavior. The mice were housed in a vivarium with 5 mice per cage (males and females were kept separate) and maintained under a 12 hr light/dark cycle. All experiments were carried out during the light cycle.

### Maze description

The maze apparatus consists of an aluminum frame, a central arena and two home boxes that can move around the arena (**Figure 1a**, **Figure 1—figure supplement 1**). The floor of the arena (a white acrylic disc 1-cm-thick and 95 cm wide) is supported by a mounted bearing (UC205-16, UCFT205-16) fixed to a central rod (2.54 cm-diameter, 23 cm-high, stainless steel) and by 4 wheels at a 14.5 cm distance from the periphery. One of the wheels is motorized (NEMA 17HS4401 Bipolar Stepper Motor) to control the rotation of the platform. The angular position of the floor is calculated from the motor increment signal and an LED-photodetector couple detecting the passages of 24 beam-breakers (2x0.5 × 20 mm plastic pieces) fixed under the floor at positions matching the arena doors.

The arena is enclosed by a 16-cm-high peripheral wall made of white acrylic (thickness: 0.3 cm) with 24 evenly distributed entrances 8 cm apart from each other. Each entrance consisted of a 5 cm hole with a flapping door made from a 3D printer (ABS, Stratasys/F120).

The entrance doors are normally closed, but following the alignment with a home box, are rotated into open positions throughout the opening of the box door. The arena is partitioned by 7-cm-high internal walls made of styrofoam blocks, on which a transparent-tainted acrylic cover (50-mm-thick) is laid to prevent mice from leaving the maze and seeing the room cues while allowing mouse tracking with infrared light.

The home boxes are made with a 3D printer (ABS, Stratasys/F120) and have motorized rotating doors, allowing mouse access to the maze, and an opening on the top, allowing experimenter access to the mice. Upon entering the box, mice have to turn ~180 degree around an internal wall to reach a water dispenser, ensuring that no body parts (such as the tail) remain near the door when it closes.

An LED-photodetector couple triggers the water reward and door closure upon the mouse reaching the water dispenser.

The home boxes are coupled to the central axle of the maze through an aluminum arm and a mounted bearing. A motorized wheel fixed under the home boxes allows the home boxes to roll on an aluminum platform along the circumference of the arena. The angular position of the box is calculated from the motor increment signal and an LED-photodetector couple detecting the passages of 24 beam-breakers (2x0.5 × 20 mm plastic pieces) fixed under the aluminum platform at positions matching the arena doors.

To control the maze, a microcontroller (Arduino mega 2560) is used. The home box electronics are connected to the microcontroller through a long ribbon cable, which is placed inside a flexible plastic accordion pipe (148 cm-long, 25 mm-diameter) to prevent cable entanglement during the rotation of the boxes.

## Maze environment

For the spatial navigation task, the arena was accessed via angled vestibules, which limited the visibility of doors from within the arena (*Figure 1—figure supplement 1*). Two luminous landmarks were placed on the arena, consisting of small (WxLxH, 5x5 × 5 cm) and large (10x15 × 5 cm) semi-transparent plastic boxes containing yellow and blue LEDs, respectively. They moved together with the arena when the arena was rotated.

## Task and maze operation

The task involved navigating from a random start position to a fixed goal position (*Figure 1b*). At the beginning of each session, a mouse was placed in one of the home boxes. The start position was adjusted by rotating the arena floor until the box was aligned with the desired start position, while the goal position was set by moving the other box until it was aligned with the desired goal position. The doors of the boxes opened to let the mouse navigate the maze and then closed when the mouse reached the lick port of the other box. For most trials, the box containing the mouse remained still while the other box and arena floor were moved. However, both boxes were moved when the rotation angle of one of the boxes was going to exceed 180° to limit the twisting of electric cables. All these steps were carried out automatically using a program uploaded to a microcontroller using Arduino IDE 1.8.19.

Mice performed a total of 10 trials per session, for 19 sessions (1 session per day), with a distinct sequence of random start positions for each session. The goal position was kept the same for sessions 1–15 and then changed for sessions 16–19. The same sequence of start positions and the same goal positions were used for all mice. Start positions were randomly selected among vestibules at least 2 door-intervals away from the goal position.

## Probe test

Probe tests were carried out 3 hr after the task, on day 15. The mice were placed in the arena and allowed to explore freely for 2 min, with all arena doors closed. The maze area was divided into 24 equal zones, with the goal position at the center of one of the zones. The time spent in each zone and quadrant was measured using Ethovision (Noldus, Spink, & Tegelenbosch, 2001).

## Trajectory tracking

Animal behavior was recorded using an infrared video camera (Basler acA1300-60gm, 25 frame/s) and infrared lights (AP-XM722-WAB-U4), and synchronized with task signals from the microcontroller using the software Ethovision (Noldus, Spink, & Tegelenbosch, 2001). Mouse position tracking and further analyses were implemented in MATLAB (MathWorks Inc, MA, USA). Video frames from a period without the mouse (the period between stopping the rotations of the boxes/arena floor and the opening of the door) were used to compute an average baseline frame that was subtracted from subsequent video frames with the mouse on the arena. For each frame, pixels delineating the mouse area were detected (using a threshold) within a circular region (radius: 7 cm) centered initially on the start position and then on the previous mouse position. The mouse area was longitudinally divided into four equal segments, and the center of mass xy-coordinates were computed for each segment,

allowing a tracking of mouse position and body orientation. For each trial, the xy-coordinates were rotated to align the maze environment and goal position across trials.

## Trajectory segmentation

On each trial, mice often visited several vestibules before reaching the goal box. We divided each trial's trajectory into two distinct components: vestibule visits and running segments between vestibules. We defined vestibule entrances and exits as the points where the mice crossed a radial distance threshold of 47.5 cm from the center of the arena.

## Path length

Mouse displacement was computed for each time step based on Euclidean distance and integrated over the entire trajectory to obtain the path length. We measured path length for both individual trials and individual segments, defined as the path between two vestibules.

## Trial length

The trial length was defined as the number of segments in a trial.

## Segment length and direction

The segment length was defined as the number of doors it spanned, considering the travel direction associated with the smallest number. Each segment could span from 1 to 12 doors, either in a clockwise or counterclockwise direction (see *Figure 2a*). A positive value of segment length indicated a clockwise direction, while a negative value indicated a counterclockwise direction.

## Serial bout length

For each trial, we calculated the length of segments and identified consecutive 1-door-long segments, which we defined as serial bouts. We measured the length of serial bouts as the number of segments in the group, with a minimum length of 1 segment corresponding to single, isolated 1-door-long segments.

## Stochastic process for random strategy

To model the pattern of vestibule visits, stochastic processes were implemented to choose which of the vestibules was visited next (*Figure 4a*). For the random strategy, the identity of the next vestibule was randomly drawn from the uniform distribution in the range [1–24], using MATLAB function randi to generate uniformly distributed pseudorandom integers in the range [1–24].

## Stochastic process for spatial strategy

For the spatial strategy, the identity of the next vestibule was randomly drawn from a distribution P that exponentially decayed as a function of goal distance (number of door-intervals between the vestibule and the goal). The distribution P was characterized by two symmetrical negative exponentials exp(-x/tau) that decayed as a function of leftward/rightward goal distance. An integer was randomly drawn from the uniform distribution in the range [1 sum_of_the_distribution_P]. The goal distance was determined by the position x for which the cumulative sum of the distribution P reached the value of the integer. The decay rate (tau) of the exponentials was heuristically set to 2 through comparison of experimental and simulated data (*Figures 2 and 4*). Importantly, we assumed that the next vestibule could not be the same as the current vestibule for this goal oriented strategy. Therefore the stochastic process was run again if the current vestibule was selected as the next vestibule.

## Stochastic process for serial strategy

For the serial strategy, the identity of the next vestibule was determined by incrementing the current vestibule number by a stochastically determined step.

The serial strategy could occur in either the clockwise or counterclockwise direction. For the mixture model, a unified process integrating both directions was implemented. The sign of the step was randomly drawn, with an 80% (20%) probability of being positive (negative), which indicated a step in the clockwise (counterclockwise) direction. Specifically, an integer x was randomly drawn from the uniform distribution in the range [0 100]. If x<80, the sign was positive, otherwise it was negative.

The size of the step was obtained by adding a random jitter to an offset value (1.2 and –2 for clockwise and counterclockwise directions, respectively). The jitter was randomly drawn from the normal distribution, using MATLAB function randn to generate normally distributed pseudorandom numbers, and was multiplied by a standard deviation value (1.2 and 1.5 for clockwise and counterclockwise directions, respectively).

The values of offsets, standard deviations, and sign probabilities were heuristically determined through comparison of experimental and simulated data (*Figures 2 and 4*). Specifically, for the two peaks observed in the segment length distribution, the offsets determined the positions of the peaks, the standard deviation determined the width of the peaks, and the sign probabilities determined the ratio of the height of the peaks.

As for the spatial process, we assumed that the next vestibule could not be the same as the current vestibule, so the stochastic process was rerun if the current vestibule was selected as the next vestibule.

## Mixture model

We implemented a mixture model that combine three strategies (random, spatial and serial) to simulate the pattern of vestibule visits (*Figure 5a*). Each strategy was assigned a specific probability (P_random, P_spatial, P_serial) for a given simulation. To select the strategy, an integer x was randomly drawn from the uniform distribution in the range [0 100]. The strategy was random if x<P_random; it was serial if x>P_random +P_spatial; otherwise, the strategy was spatial.

The mixture model assumed that mice use the same strategy throughout a specific number N of consecutive vestibules. At the beginning of a trial, a strategy was randomly selected, and its associated stochastic process was recursively used to draw the next N vestibules. This process was repeated: the strategy was randomly selected again to draw the next N vestibules until the goal was reached.

We ran simulations of the mixture model for the same number of mice and trials as in the experiments, and used the same start positions as in the experiments.

## Estimation of strategy proportion and number N of consecutive vestibules for the mixture model

To determine the optimal probability values for each strategy (P_random, P_spatial, P_serial), we conducted a comprehensive simulation study, exploring a wide range of probability combinations. Specifically, we tested values of P_random, P_spatial, and P_serial ranging from 0 to 100 (percent) with an increment of 2, while ensuring that their sum was always equal to 100. For each combination of probabilities, we used the same analytical methods as in the experimental analysis and computed the mean square error between the simulated and experimental distributions. We selected the optimal probability values as those that produced the lowest mean square error between the simulated and experimental distributions.

To determine P_random, P_spatial and P_serial for each of the 19 days, all simulations and error calculations were repeated for each day, using the same number of mice, trials and the same start positions as in the experiments.

To determine the optimal number N of consecutive vestibules, we repeated all simulations and error calculations for values of N ranging from 1 to 15 and selected the value associated with the lowest mean square error between the simulated and experimental distributions (*Figure 5c*).

## Markov chain model

The model incorporated four strategies: random, spatial, serial clockwise and serial counterclockwise. To generate vestibule sequences, the model iteratively draw a strategy and then a vestibule using the stochastic process associated with the selected strategy, repeating this operation until the goal (vestibule 0) was reached. At the beginning of the trials, the strategy was drawn according to a set of probabilities determining the initial strategy (*Figure 6a*, left). Otherwise, the model transitioned between strategies based on another set of probabilities (*Figure 6a*, right).

## Genetic algorithm

A genetic algorithm was employed to estimate the probabilities for both the initial strategy and strategy transitions in the Markov chain model. It iteratively refined probability values to minimize mean square errors between experimental and simulated distributions.

The algorithm operated with a population size of 500 individuals, with each individual representing a 5x4 probability matrix. The first row stored the probabilities for the initial strategy, while the subsequent rows stored the probabilities for strategy transitions. Initial values in individuals' probability matrices were randomly assigned.

The algorithm ran for 500 generations. In each generation, vestibule sequences and mean square errors between experimental and simulated distributions were computed for each individual. The 250 individuals with the lowest mean square errors were selected as the 'best' and passed to the next generation. The next generation was complemented by a new set of 250 individuals, formed by either mutating the best individuals or mixing elements from pairs of the best individuals. Mutation and mixing operations alternated in every other generation.

Mutation involved adding random values (−0.1–0.1) to each element of the probability matrix. Negative values were set to 0, and normalization ensured probabilities remained in the range of 0–1 and summed to 1 across strategies. Mixing elements from pairs of individuals involved combining randomly selected subsets of rows from one individual with complementary subsets from another individual.

The final probability matrix was obtained by averaging the matrices of the 250 best individuals from the 500th generation. The reliability of the results was assessed by observing similar convergence of parameters across 10 repetitions of the algorithm.

## Statistical analysis

All statistical analyses were performed in Matlab (MathWorks). Number of animals were similar to those generally employed. For each distribution, a Kolmogorov-Smirnov test was used to test the null hypothesis that the sample distribution was derived from a standard normal distribution. If normality was uncertain, we used non-parametric tests as stated in the main text or figures. Otherwise, Student t-tests were used to test the sample mean. Correlations were computed using Pearson's correlation coefficient.

## Acknowledgements

This work was supported by the Korea Institute of Science and Technology Institutional Program (Project # 2E32211) and the National Research Foundation of Korea (NRF Grant # 2021R1A2C3005560).

## Additional information

### Funding

| Funder | Grant reference number | Author |
|---|---|---|
| National Research Foundation of Korea | 2021R1A2C3005560 | Sebastien Royer |
| Korea Institute of Science and Technology | 2E32211 | Sebastien Royer |

The funders had no role in study design, data collection and interpretation, or the decision to submit the work for publication.

### Author contributions

Ju-Young Lee, Conceptualization, Data curation, Formal analysis, Investigation, Methodology, Writing – original draft, Writing – review and editing; Dahee Jung, Methodology, Writing – review and editing; Sebastien Royer, Conceptualization, Data curation, Formal analysis, Supervision, Funding acquisition, Investigation, Methodology, Writing – original draft, Writing – review and editing

## Author ORCIDs
Sebastien Royer https://orcid.org/0000-0002-3038-1129

## Ethics

All experiments were conducted in accordance with institutional regulations (Institutional Animal Care and Use Committee of the Korea Institute of Science and Technology) and conformed to the Guide for the Care and Use of Laboratory Animals (NRC 2011).

Reviewer #1 (Public Review) https://doi.org/10.7554/eLife.88648.4.sa1
Reviewer #2 (Public Review): https://doi.org/10.7554/eLife.88648.4.sa2
Reviewer #3 (Public Review): https://doi.org/10.7554/eLife.88648.4.sa3
Author response https://doi.org/10.7554/eLife.88648.4.sa4

## Additional files

### Supplementary files
• MDAR checklist

### Data availability

The data that were collected for this study, as well as all the codes for data analysis and modeling are available at GitHub (copy archived at *Royer, 2024*).

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
