## [Editor Report · eLife assessment]

This study presents a **valuable** new behavioral apparatus aimed at differentiating the strategies animals use to orient themselves in an environment. The evidence supporting the claims is **solid**, with statistical modeling of animal behavior. Overall, this study will attract the interest of researchers exploring spatial learning and memory.

---

## [Referee Report · Reviewer #1 (Public Review)]

The authors design an automated 24-well Barnes maze with 2 orienting cues inside the maze, then model what strategies the mice use to reach the goal location across multiple days of learning. They consider a set of models and conclude that the animals begin with a large proportion of random choices (choices irrespective of the goal location), which over days of experience becomes a combination of spatial choices (choices targeted around the goal location) and serial choices (successive stepwise choices in a given direction). Moreover, the authors show that after the animal has many days of experience in the maze, they still often began each trial with a random choice, followed by spatial or serial choices.

This study is written concisely and the results are presented concisely. The best fit model provides valuable insight into how the animals solve this task, and therefore offers a quantitative foundation upon which tests of neural mechanisms of the components of the behavioral strategy can be performed. These tests will also benefit from the automated nature of the task.

---

## [Referee Report · Reviewer #2 (Public Review)]

This paper uses a novel maze design to explore mouse navigation behaviour in an automated analogue of the Barnes maze. A major strength is the novel and clever experimental design which rotates the floor and intramaze cues before the start of each new trial, allowing the previous goal location to become the next starting position. The modelling sampling a Markov chain of navigation strategies is elegant, appropriate and solid, appearing to capture the behavioural data well. This work provides a valuable contribution and I'm excited to see further developments, such as neural correlates of the different strategies and switches between them.

---

## [Referee Report · Reviewer #3 (Public Review)]

The development of an automated Barnes maze allows for more naturalistic and uninterrupted behavior, facilitating the study of spatial learning and memory, as well as the analysis of the brain's neural networks during behavior when combined with neurophysiological techniques. The system's design has been thoughtfully considered, encompassing numerous intricate details. These details include the incorporation of flexible options for selecting start, goal, and proximal landmark positions, the inclusion of a rotating platform to prevent the accumulation of olfactory cues, and careful attention given to atomization, taking into account specific considerations such as the rotation of the maze without causing wire shortage or breakage. When combined with neurophysiological manipulations or recordings, the system provides a powerful tool for studying spatial navigation system.

The behavioral experiment protocols, along with the analysis of animal behavior, are conducted with care, and the development of behavioral modeling to capture the animal's search strategy is thoughtfully executed. It is intriguing to observe how the integration of these innovative stochastic models can elucidate the evolution of mice's search strategy within a variant of the Barnes maze.

---

## [Author Response]

The following is the authors’ response to the previous reviews.

**Public Reviews:**

**Reviewer #1 (Public Review):**
The authors design an automated 24-well Barnes maze with 2 orienting cues inside the maze, then model what strategies the mice use to reach the goal location across multiple days of learning. They consider a set of models and conclude that the animals begin with a large proportion of random choices (choices irrespective of the goal location), which over days of experience becomes a combination of spatial choices (choices targeted around the goal location) and serial choices (successive stepwise choices in a given direction). Moreover, the authors show that after the animal has many days of experience in the maze, they still often began each trial with a random choice, followed by spatial or serial choices.This study is written concisely and the results are presented concisely. The best fit model provides valuable insight into how the animals solve this task, and therefore offers a quantitative foundation upon which tests of neural mechanisms of the components of the behavioral strategy can be performed. These tests will also benefit from the automated nature of the task.
**Reviewer #2 (Public Review):**
This paper uses a novel maze design to explore mouse navigation behaviour in an automated analogue of the Barnes maze. A major strength is the novel and clever experimental design which rotates the floor and intramaze cues before the start of each new trial, allowing the previous goal location to become the next starting position. The modelling sampling a Markov chain of navigation strategies is elegant, appropriate and solid, appearing to capture the behavioural data well. This work provides a valuable contribution and I'm excited to see further developments, such as neural correlates of the different strategies and switches between them.
**Reviewer #3 (Public Review):**
Strength:The development of an automated Barnes maze allows for more naturalistic and uninterrupted behavior, facilitating the study of spatial learning and memory, as well as the analysis of the brain's neural networks during behavior when combined with neurophysiological techniques. The system's design has been thoughtfully considered, encompassing numerous intricate details. These details include the incorporation of flexible options for selecting start, goal, and proximal landmark positions, the inclusion of a rotating platform to prevent the accumulation of olfactory cues, and careful attention given to atomization, taking into account specific considerations such as the rotation of the maze without causing wire shortage or breakage. When combined with neurophysiological manipulations or recordings, the system provides a powerful tool for studying spatial navigation system.The behavioral experiment protocols, along with the analysis of animal behavior, are conducted with care, and the development of behavioral modeling to capture the animal's search strategy is thoughtfully executed. It is intriguing to observe how the integration of these innovative stochastic models can elucidate the evolution of mice's search strategy within a variant of the Barnes maze.Comments on revised version:The authors have addressed all the points I outlined in the previous round of review, resulting in significant improvements to the manuscript. However, I have one remaining comment. Given the updated inter-animal analysis (Supplementary Figure 8), it appears that male and female mice develop strategies differently across days. Male mice seem to progressively increase their employment of spatial strategy across days, at the expense of the random strategy. Conversely, female mice exhibit both spatial and serial strategies at their highest levels on day 2, with minimal changes observed on the subsequent days.These findings could alter the interpretation of Figure 5 and the corresponding text in the section "Evolution of search strategy across days".For instance, this statement on page 6 doesn't hold for female mice: "The spatial strategy was increased across days, ... largely at the expense of the random strategy."

We agree with the reviewer. While the text on page 6 is still valid for the male-female pooled data, we have clarified in the next section describing male-female differences that this trend is not observed in female. Furthermore, we adjusted the relevant part of the discussion the following manner:

“A shift in the proportion of random, spatial and serial strategies was observed across days. Several factors might contribute to this shift, including learning of the environment and goal location, changes in motivation for exploration versus goal-directed navigation, and the evaluation of each strategy’s benefit via reinforcement learning. The spatial strategy progressively increased, mostly at the expense of the random strategy. This trend suggests a diminishing interest in exploration and an increasing benefit from employing the spatial strategy as the mice became more familiar with the environment and goal location. Consistent with this hypothesis, the development of the spatial strategy approximately matched the development of spatial maps in the hippocampus37 and the growth pattern of hippocampal feedforward inhibitory connectivity62, both showing progressive increases that reached plateaus after a week. In contrast, the serial strategy showed a sudden increase from day 1 to day 2, indicating that this goal-directed strategy is associated with rapid learning and could already be reinforced on day 2. However, the strategy shift was not uniform across the mouse population, as male and female mice showed distinct trends. Female mice showed no progressive increase in spatial strategy and initially relied more on the spatial strategy while using the random strategy less compared to male mice. This difference might be explained by faster learning of goal location and/or a stronger inclination towards goal-directed navigation over exploration in female mice.”

**Recommendations for the authors:**

**Reviewer #1 (Recommendations For The Authors):**
Minor points:(1) The following sentence in the abstract is not grammatical: "The processes randomly selected vestibules based on either uniform (random) or biased (serial and spatial) probability distributions; closely matched experimental data across a range of statistical distributions characterizing the length, distribution, step size, direction, and stereotypy of vestibule sequences; and revealed a shift from random to spatial and serial strategies over time, with a strategy switch occurring approximately every 6 vestibule visits."One possible revision is: "The processes randomly selected vestibules based on either uniform (random) or biased (serial and spatial) probability distributions; [they] closely matched experimental data across a range of statistical distributions characterizing the length, distribution, step size, direction, and stereotypy of vestibule sequences, [revealing] a shift from random to spatial and serial strategies over time, with a strategy switch occurring approximately every 6 vestibule visits."

We followed the reviewer’s suggestion.

(2) There is a missing word in the following sentence in the last paragraph of the discussion: "Our tools might be combined in the future with optogenetic and/or pharmacogenetic [missing word here] to investigate the neural mechanisms underlying strategy selection"

We added the word ‘manipulations’: ‘… optogenetic, pharmacogenetic manipulations …’

**Reviewer #2 (Recommendations For The Authors):**
I have two minor suggestions:(1) Results - Automated Maze section: It would be beneficial to clarify here that the floor and cues rotate allowing automation by chining start/end positions together. This information is key to the reader understanding the task and currently they would only know this by studying fig1 or delving into the methods

As suggested by the reviewer, we have added the following text in the Results - Automated Maze section:

“The maze consist of an enclosed arena with an array of 24 doors evenly spaced along the periphery, and two home boxes moving around the arena perimeter. Start positions are changed by rotating the arena and the home boxes (Fig. 1b). Furthermore, the arena has a tinted cover that prevents mice from seeing room cues while still allowing for infrared tracking of mouse trajectories.”

(2) I still find the author's decision to exclude days from some of the line plots, e.g. days 3,4,5 from Fig2 etc, a little odd as this makes the reader wary. I appreciate their argument about clarity, but this can still be achieved while partitioning all of the data rather than excluding certain days. NB I do not find the heat map distributions in the far panel a particularly good substitute for this as pixel intensities are far less interpretable

We appreciate the reviewer’s comment. We want to point out that line plots for all individual days are actually displayed in Supplementary Figure 7a.

**Reviewer #3 (Recommendations For The Authors):**
Although the difference between females and males is clear in Figure S8b, please note that the statistics in panels C and D might not be appropriate, as many of them may become insignificant if adjusted for multiple comparisons.

If we understand correctly, a Bonferroni correction would need to consider the 3 day intervals in Figure S8c and the 2 day groups in Figure S8d. This would mean a significance threshold of 0.05/3 = 0.016667 for Figure S8c and 0.05/2 = 0.025 for Figure S8d, after Bonferroni correction. As it stands, all comparisons that are not labelled ’ns’ in Figure S8c-d remain significant even after applying the Bonferroni correction.